# Battling the obesity epidemic with a school-based intervention: Long-term effects of a quasi-experimental study

M. Willeboordse[1]*, N. H. M. Bartelink[2], P. van Assema[2], S. P. J. Kremers[3], H. H. C. M. Savelberg[4], M. T. H. Hahnraths[1], L. Vonk[5,6], M. Oosterhoff[7¤], C. P. van Schayck[1], B. Winkens[8], M. W. J. Jansen[5,6]

1 Department of Family Medicine, Care and Public Health Research Institute (CAPHRI), Maastricht University, Maastricht, The Netherlands, 2 Department of Health Promotion, Care and Public Health Research Institute (CAPHRI), School of Nutrition and Translational Research in Metabolism (NUTRIM), Maastricht University, Maastricht, The Netherlands, 3 Department of Health Promotion, School of Nutrition and Translational Research in Metabolism (NUTRIM), Maastricht University, Maastricht, The Netherlands, 4 Department of Nutrition and Movement Sciences, School of Nutrition and Translational Research in Metabolism (NUTRIM) and School of Health Professions Education (SHE), Maastricht University, Maastricht, The Netherlands, 5 Department of Health Services Research, Care and Public Health Research Institute (CAPHRI), Maastricht University, Maastricht, The Netherlands, 6 Academic Collaborative Centre for Public Health Limburg, Heerlen, The Netherlands, 7 Department of Clinical Epidemiology and Medical Technology Assessment, Care and Public Health Research Institute (CAPHRI), Maastricht University Medical Centre, Maastricht, The Netherlands, 8 Department of Methodology and Statistics, Care and Public Health Research Institute (CAPHRI), Maastricht University, Maastricht, The Netherlands

¤ Current address: Centre for Nutrition, Prevention and Health Services, National Institute for Public Health and the Environment (RIVM), Bilthoven, The Netherlands
* maartje.willeboordse@maastrichtuniversity.nl

**Data Availability Statement:** The underlying database of this article has been uploaded and is openly available in the data warehouse of

## Abstract

### Background

School-based health-promoting interventions are increasingly seen as an effective population strategy to improve health and prevent obesity. Evidence on the long-term effectiveness of school-based interventions is scarce. This study investigates the four-year effectiveness of the school-based Healthy Primary School of the Future (HPSF) intervention on children's body mass index z-score (BMIz), and on the secondary outcomes waist circumference (WC), dietary and physical activity (PA) behaviours.

### Methods and findings

This study has a quasi-experimental design with four intervention schools, i.e., two full HPSFs (focus: diet and PA), two partial HPSFs (focus: PA), and four control schools. Primary school children (aged 4–12 years) attending the eight participating schools were invited to enrol in the study between 2015 and 2019. Annual measurements consisted of children's anthropometry (weight, height and waist circumference), dietary behaviours (child- and parent-reported questionnaires) and PA levels (accelerometers). Between 2015 and 2019, 2236 children enrolled. The average exposure to the school condition was 2·66 (SD 1·33) years, and 900 participants were exposed for the full four years (40·3%). After

Maastricht University (see https://dataverse.nl/dataset.xhtml?persistentId=doi:10.34894/ZLB24E). The link to this database has been added to the manuscript.

**Funding:** This study was funded by the Limburg provincial authorities (www.limburg.nl), Project Number 200130003 (received by CvS), by Maastricht University (www.maastrichtuniversity.nl), Project Number 200130003 (received by MJ) and by FrieslandCampina (www.frieslandcampina.nl) (received by MW), Project Number LLMV00. The funders had no role in study design, data collection and analysis, decision to publish, or preparation of the manuscript.

**Competing interests:** The authors have declared that no competing interests exist.

four years of intervention, both full (estimated intervention effect (B = -0·17 (95%CI -0·27 to -0·08) p = 0·000) and partial HPSF (B = -0·16 (95%CI-0·25 to -0·06) p = 0·001) resulted in significant changes in children's BMIz compared to control schools. Likewise, WC changed in favour of both full and partial HPSFs. In full HPSFs, almost all dietary behaviours changed significantly in the short term. In the long term, only consumption of water and dairy remained significant compared to control schools. In both partial and full HPSFs, changes in PA behaviours were mostly absent.

## Interpretation

This school-based health-promoting intervention is effective in bringing unfavourable changes in body composition to a halt in both the short and long term. It provides policy makers with robust evidence to sustainably implement these interventions in school-based routine.

## Introduction

The ongoing rise in childhood obesity levels implies that not intervening will lead to a growth of the obesity epidemic [1]. From a societal perspective, health promotion efforts targeting young children can be highly effective, as healthy habits learned at a young age often track into adult life and thereby have the potential to induce life-long effects on chronical conditions, health care costs and societal participation [2–4]. Targeting children via schools is increasingly seen as an effective population strategy to improve health, which resulted in the Health Promoting School approach developed by the World Health Organisation (WHO). Schools are an ideal environment for public health initiatives, as schools enable multiple aspects of health promotion to be modified, while reaching children from a variety of socioeconomic and ethnic backgrounds. Current evidence suggests that school-based interventions show small favourable effects in terms of body mass index (BMI), and dietary and physical activity (PA) behaviours [5–9]. In most studies, effects on body composition are measured using BMI [5]. BMI alone might not correctly reflect effects of PA interventions, as PA could increase lean body mass without changing BMI [6]. Therefore, there is a need for studies including both indirect (e.g. BMI) and more direct outcome measures of adiposity (e.g. waist circumference (WC)).

School-based interventions can only be of societal and clinical relevance if intervention effects are implemented sustainably and maintained for a prolonged time [10]. Yet, prolonged (>24 months) studies with sufficient power are scarce (S1 Appendix). Although no consensus exists, the most promising interventions often adopt a whole-school approach to health with combined intervention components (e.g. PA and diet) and high parental involvement [7, 8, 11]. Embracing the current evidence on school-based intervention programmes, the Healthy Primary School of the Future (HPSF) was developed in the Netherlands in 2015. The design of the study allows investigation of the effects of long-term exposure (four years) to the HPSF intervention, which is split into a full and a partial version. The full HPSF focusses on diet and PA and the partial HPSF focusses only on PA. All activities are supervised by professional staff during prolonged schooltime [12]. In the short-term evaluation (one and two years), it was shown that both full and partial HPSFs led to several significant improvements in children [13, 14]. The greatest effects were found in children's water consumption at school and their lunch intake, while smaller effects were observed in children's PA behaviours and their overall

dietary behaviours. After one and two years, a small favourable effect on children's body mass index z-score (BMIz) was found. In the current paper, we present the effects of long-term (i.e. four years) exposure to the HPSF. The following research question was formulated: what is the effectiveness of long-term exposure to HPSF on children's BMIz, waist circumference (WC) and their dietary and PA behaviours, compared to children of control schools?

We hypothesise that improvements in dietary and PA behaviours will be visible after one year of exposure, and remain sustained. We hypothesise increasing intervention effects on BMIz and WC with increasing intervention exposure.

## Methods

Previous studies reported in detail on the study design and power calculation [12], process evaluation [15], non-response and external validity [16], two-year effectiveness [13, 14], and cost-effectiveness [17].

### Study design

The current study has a longitudinal quasi-experimental study design with a dynamic open cohort, in which children of two full HPSF schools and two partial HPSF schools were compared with four control schools for four years (Fig 1). All enrolled pupils (aged 4–13 years) were invited to engage in the study between 2015 and 2019, including pupils who had just entered school at four years old, or any moment later in their school career. Children were excluded after they had left school. The amount of time children were exposed to HPSF varies from 0–4 years and is expressed by the variable exposure (E0 to E4). The baseline exposure (E0) is defined by the first moment children were exposed to the school environment starting from September 2015, which is not necessarily equal to the first measurement children attended.

The need for Medical Ethical approval has been waived by the Zuyderland Medical Ethics Committee in Heerlen (METCZ:14N-142). Children were allowed to participate in the measurements if their parents or guardians signed an informed consent. Measurements were conducted annually from 2015 until 2019 in the autumn period. HPSF intervention started in November 2015 after completion of the first measurement (T0). All children in the intervention schools were exposed to the interventions, regardless of their enrolment in the scientific study. The four control schools maintained their regular school curriculum and school hours throughout the study period. During lunchtime, children in control schools consumed their lunch brought from home and engaged in free play supervised by volunteers, as is common in Dutch schools.

### Study population

The participating schools are situated in the Parkstad region in the southern part of the Netherlands. Compared to other areas in the Netherlands, Parkstad is a moderate-to-low socioeconomic area characterised by a high prevalence of chronic diseases including obesity, and a lower life expectancy than the Dutch average [18]. Schools were recruited based on voluntary participation, and the possibility of including a minimal n = 100 children per school to meet the power calculation [12]. Children and their parents were recruited using information brochures and classroom visits. If participants switched to other schools during the intervention period, only data of the original school were used. Control schools maintained the school curriculum that is currently common practice in the Netherlands.

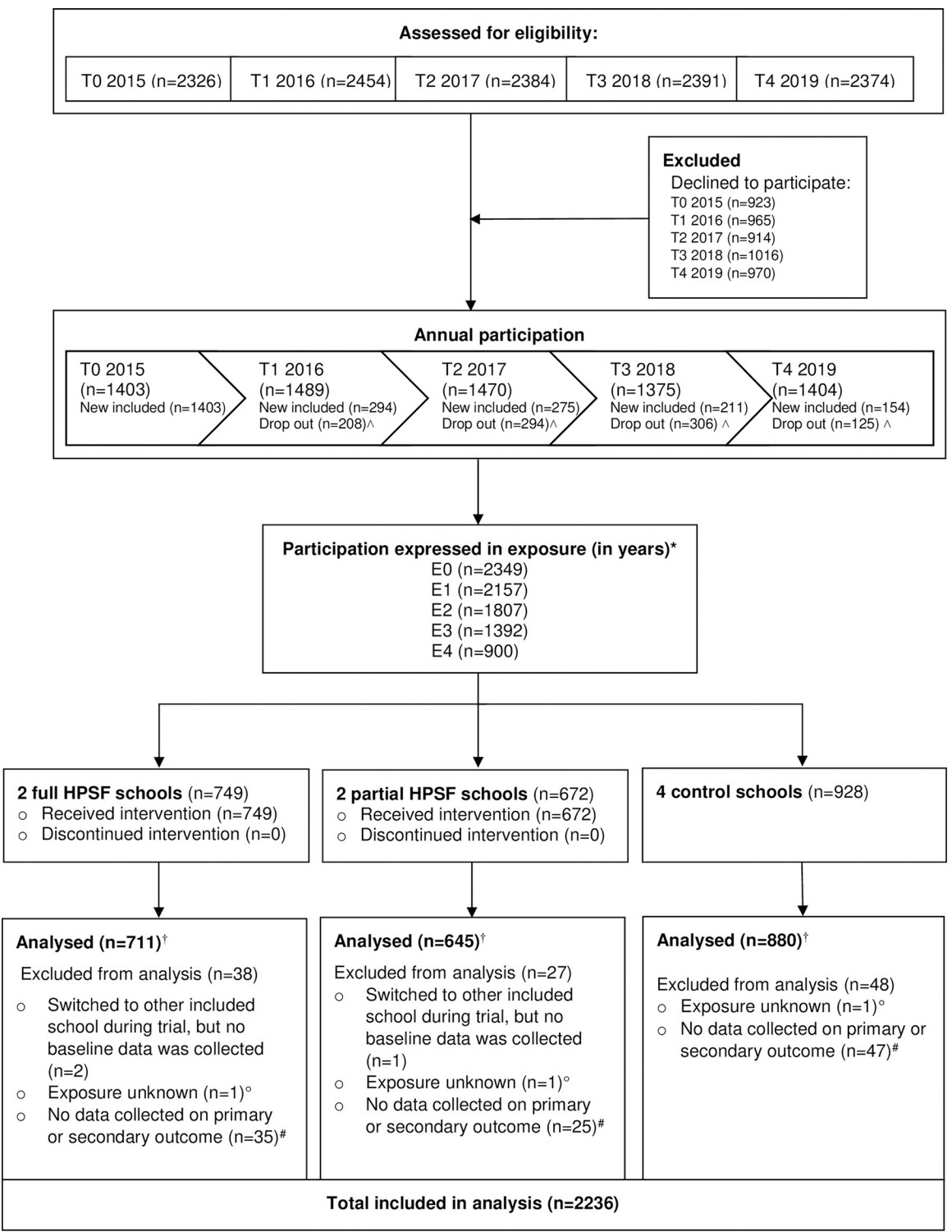

**Fig 1. Trial profile.** Abbreviations: HPSF, Healthy Primary School of the Future. ^ Reasons for drop out over the four years include graduation (n = 637), or other (n = 296) including migration, actively stopping participation or switching to other included schools. * Exposure was defined by the time (in years) children are exposed to the allocated school condition, from 2015 onwards in full years. Participants with 0 years exposure enrolled in the intervention/control condition during: T0 (n = 1699), T1 (n = 233), T2 (n = 200), T3 (n = 91), and T4 (n = 25). Participants with 1 year exposure enrolled in the intervention/control condition during T0 (n = 1553), T1 (n = 309), T2 (n = 200), and T3 (n = 90). Participants with 2 years exposure enrolled in the intervention/control condition during T0 (n = 1320), T1 (n = 287) and T2 (n = 196). Participants with 3 years exposure enrolled in the intervention/control condition during T0

(n = 1120) and T1 (n = 269). Participants with 4 years exposure enrolled in the intervention/control condition during T0 (n = 900). The children which were excluded for analyses were not included in these numbers (n = 101 with E0, n = 5 with E1, n = 4 with E2, n = 3 with E1, and n = 0 with E4). † All participants were included in the analyses. Due to the dynamic cohort design, not all students were exposed for four years. In full HPSFs, n = 510 (68·1%) students were not exposed for four years; 257 (50·4%) graduated before four years exposure, 198 (38·8%) enrolled at the school later than 2015, 39 (7·6%) migrated, 8 (1·6%) switched to other included schools and 8 (1·6%) actively stopped participation. In partial HPSFs, n = 380 (56·5%) students were not exposed for four years; 212 (55·8%) graduated, 118 (31·1%) enrolled at the school later than 2015, 29 (7·6%) migrated, 4 (1·1%) switched to other included schools, 17 (4·5%) actively stopped participation. In control schools n = 556 (59·9%) students were not exposed for four years; 320 (57·6%) graduated, 141 (25·4%) enrolled at the school later than 2015, 64 (11·5%) migrated, 3 (0·5%) switched to other included schools and 28 (5·0%) actively stopped participation. Exposure was unknown for 3 children who actively stopped participation, as schools were not able to provide us with information on school starting date due to GDPR regulations. # Reasons why no data was collected included: no data collection in children of group 1 in line with study protocol (S2 Appendix), no data collection in selected group during T3 in line with study protocol (S2 Appendix), school absence of child during measurements, incomplete parent questionnaire.

## The Healthy Primary School of the Future (HPSF)

Interventions were implemented for four years by school staff and pedagogical employees following a systematic health promotion approach based on the PRECEDE-PROCEED model and the Intervention Mapping protocol, and sustainably integrated into the whole school system based on the socioecological approach to health promotion of the WHO (Table 1). Changes were contextualised bottom-up, resulting in tailored intervention programmes, which remained largely unchanged over the four years.

## Data collection and outcomes

Data was gathered annually in schools during one week of measurements between 2015 and 2019 by non-blinded trained researchers, including anthropometry, accelerometry, questionnaires for children (on paper) and parents (digitally) [12]. The primary outcome BMIz was derived using Dutch reference values [19] Secondary outcomes included waist circumference, PA behaviours and dietary behaviours. In S2 Appendix, data collection methods and outcomes are described in detail.

## Data analysis

Initial descriptive statistics and frequencies were generated to summarise the data using IBM SPSS statistics for Windows (V.25.0; Armonk, NY, USA). Chi-square, ANOVA and Welch tests with post-hoc tests were performed to analyse baseline differences between the three conditions. All analyses are based on an intention-to-treat principle; participants are included if

**Table 1. Interventions in the Healthy Primary School of the Future (HPSF).**

| | 2 Full HPSF | 2 Partial HPSF | 2 Control schools |
|---|---|---|---|
| **Top-down initiated changes** | Free healthy mid-morning snack and lunch | - | - |
| | Structured PA sessions during lunch break* | Structured PA sessions during lunch break* | - |
| | Prolonged lunch break of 45–75 minutes | Prolonged lunch break in one school of 15 minutes | - |
| **Bottom-up initiated health-promoting interventions#** | Schools were motivated to implement interventions by trained regional youth health-promoting employees | Schools were motivated to implement interventions by trained regional youth health-promoting employees | Schools received no additional support to implement interventions |

* PA sessions were organized both in- and outdoors and alternated with cultural activities or free play for a maximum of two times per week.

# Bottom-up interventions included, amongst other things, health education lessons, free water bottle distribution, vegetable gardens, policy regarding food, beverages, birthday treats and festivities, active school transportation, EU-funded fruit and vegetables scheme, energizer breaks, additional physical education or swimming lessons, additional interventions for overweight and obese children.

they completed at least one annual measurement round. Two-level linear mixed-model analyses were used to analyse continuous outcome measures, General Estimating Equations (GEE) were used for the binary outcome variable (lunch intake). Repeated measurements were included as first level and participants as second level. The best fitting covariance structure was determined for each analysis, based on the lowest Bayesian Information Criterion (BIC) value. Condition (full HPSF, partial HPSF, and control), exposure (categorical, ranging from E0-E4) and the interaction term condition*exposure were included as fixed factors. The following covariates were included: sex, age in years (at E0), socioeconomic status (in tertiles), ethnicity (Western/non-Western), BMIz (at E0; only for the behavioural outcomes), and weather variables for the analyses regarding children's PA behaviours (sun in hours, precipitation in mm, and mean temperature in degrees Celsius). In all analyses adjustments were made for baseline values of the outcome variable. Missing covariates were imputed using multiple imputation techniques with fully conditional specification and 10 iterations, generating 50 complete datasets. Missing outcome variables were handled using a likelihood-based approach, as is common in mixed-model analyses [20]. Analyses with dietary variables as outcome measures were only performed for full HPSFs.

Following our initial hypothesis, the outcome measures BMIz and WC are presented as trend analyses. A linear trend of exposure was assessed by considering exposure as a numerical variable, after which pooled estimates were calculated over a four-year trend rather than separate annual effects. Linearity assumption was checked by adding and testing a quadratic exposure term and its interaction with condition in addition to visually checking the trend using the annual effects.

For the primary outcome BMIz, results were stratified by sex, and two sensitivity checks were performed: a 'closed-cohort analysis' with only children with four years exposure included, and an analysis with BMI in kg/m$^2$ as outcome measure.

For all tests, a p-value $\leq 0.05$ was considered statistically significant. Standardised effect sizes (ES) were determined for continuous outcome variables, computed as the pooled estimated mean difference divided by the square root of the pooled residual variance at baseline. Categorical outcomes resulted in odds ratios (OR).

This trial was registered at Clinicaltrials.gov (NCT02800616) in June 2016.

## Results

2236 children were included in the current study. Due to the dynamic cohort design, not all children were exposed to the school condition for the full four years. The average exposure was 2·66 (SD 1·33) years. N = 2157 children were exposed for minimally one year (96·5%), and 900 participants for the full four years (40·3%) (Fig 1). The majority (n = 1699, 76·0%) of the children were first exposed (E0) at the start in 2015 (T0). Out of all students enrolled, annual study participation ranged from 57·7–61·7%, with relatively more children participating in intervention schools than in control schools. At baseline, children in control schools showed less favourable dietary and PA behaviours, were more often overweight and obese, and had a lower socioeconomic status compared with the full and partial HPSFs (Table 2). The average annual response rate for the different measurement instruments varied between 46·2% and 95·0% (child questionnaire 95·0%, lunch questionnaire 92·1%, anthropometrics 92·6%, accelerometry 60·8%, parental questionnaire 46·2%).

### Primary outcome: Children's BMIz (Table 3, Fig 2, S3, S4 Appendices)

BMIz in both full and partial HPSFs remained unchanged over time, whereas BMIz of children in the control schools increased significantly over time, as shown in Fig 2. Compared to

**Table 2. Characteristics of the participants at baseline (E0).**

| | Total (n = 2236) | | Full HPSF (n = 711) | | Partial HPSF (n = 645) | | Control (n = 880) | |
|---|---|---|---|---|---|---|---|---|
| | N | %/ mean (SD) | N | %/ mean (SD) | N | %/ mean (SD) | N | %/ mean (SD) |
| Sex (% boys) | 2236 | 47·9 | 711 | 48·4 | 645 | 48·7 | 880 | 46·8 |
| Age (years) | 2236 | 7·19 (2·48) | 711 | 7·24 (2·47) | 645 | 7·08 (2·53) | 880 | 7·22 (2·46) |
| Ethnicity (% Western) | 1438 | 93·2 | 477 | 92·9 | 452 | 94·9 | 509 | 91·9 |
| Socioeconomic status (%) | 1431 | | 460 | | 466 | | 505 | |
| *Lowest tertile* | | 33·3 | | 27·6 | | 31·3 | | 40·2 |
| *Middle tertile* | | 33·2 | | 33·5 | | 35·8 | | 30·5 |
| *Highest tertile* | | 33·5 | | 38·9 | | 32·8 | | 29·3 |
| BMIz | 1302 | 0·13 (1·02) | 431 | 0·04 (0·99) | 380 | 0·12 (0·97) | 491 | 0·21 (1·07) |
| Overweight/obese (%) | 1302 | 19·5 | 70 | 16·2 | 72 | 18·9 | 112 | 22·8 |
| WC in cm | 1303 | 60·38 (8·54) | 431 | 59·83 (8·10) | 380 | 60·01 (8·11) | 492 | 61·15 (9·17) |
| Light PA (%) | 987 | 31·52 (5·58) | 340 | 31·71 (5·62) | 302 | 31·88 (5·65) | 345 | 31·02 (5·46) |
| MVPA (%) | 987 | 7·64 (2·67) | 340 | 8·19 (2·77) | 302 | 7·23 (2·55) | 345 | 7·47 (2·58) |
| Sedentary behaviour (%) | 987 | 60·84 (7·03) | 340 | 60·11 (7·17) | 302 | 60·89 (7·04) | 345 | 61·51 (6·82) |
| Healthy dietary behaviours (mean days/week)[1] | 977 | 5·13 (1·08) | 340 | 5·25 (1·10) | 331 | 5·19 (1·01) | 306 | 4·93 (1·10) |
| Unhealthy dietary behaviours (mean days/week)[2] | 971 | 1·11 (0·68) | 338 | 1·09 (0·69) | 329 | 1·04 (0·61) | 304 | 1·21 (0·73) |
| School water consumption (0–3)[3] | 908 | 1·29 (1·14) | 307 | 1·52 (1·78) | 244 | 1·38 (1·22) | 357 | 1·04 (1·00) |
| Fruit at lunch (% yes) | 1048 | 38·1 | 345 | 45·5 | 304 | 38·5 | 399 | 31·3 |
| Vegetables at lunch (% yes)[1] | 1050 | 25·2 | 347 | 32·6 | 303 | 25·1 | 400 | 19·0 |
| Grains at lunch (% yes)[4] | 1055 | 87·0 | 348 | 92·5 | 305 | 88·5 | 402 | 81·1 |
| Dairy at lunch (% yes)[5] | 1043 | 40·8 | 342 | 47·4 | 302 | 39·7 | 399 | 36·1 |
| Water at lunch (% yes) | 1043 | 34·0 | 343 | 37·3 | 302 | 41·1 | 398 | 25·9 |
| Butter at lunch (% yes) | 1055 | 48·2 | 347 | 48·1 | 307 | 54·4 | 401 | 43·6 |
| Minimum of two healthy food groups at lunch (% yes)[6] | 1063 | 81·9 | 353 | 84·7 | 307 | 84·7 | 403 | 77·4 |

[1] Healthy dietary behaviours is a composite score for frequency of consumption of breakfast, fruits, vegetables, and water.

[2] Unhealthy dietary behaviours is a composite score for frequency of consumption of soft drinks, sport drinks, energy drinks, chocolate, salted snacks, cookies, and soft ice cream.

[3] School water consumption ranges from never (0) to daily (3).

[4] Grains consists of the items: bread and cereals.

[5] Dairy consists of the items: milk/yoghurt and cheese.

[6] Items in the healthy food groups include: fruits, vegetables, grains, dairy, water and butter.

Abbreviations: HPSF, Healthy Primary School of the Future; SD, standard deviation; BMI, body mass index; MVPA, moderate to vigorous physical activity; PA, physical activity; WC, waist circumference.

control schools, the negative trend for full HPSFs (E1: B = -0·04 (95%CI-0·07 to -0·02); E4: B = -0·17 (95%CI-0·27 to -0·08), p<0·000) and partial HPSFs (E1: B = -0·04 (95%CI-0·06 to -0·02); E4: B = -0·16 (95%CI-0·25 to -0·06), p = 0·001) on BMIz was significant from E1 to E4. This resulted at E4 in effect sizes of respectively -0·17 for full HPSF and -0·16 for partial HPSFs. The majority of children maintained their weight category after one (89·0%) and four years exposure (76·4%). The number of children developing underweight, or overweight/obesity did not differ significantly between the three conditions at either E1 (p = 0·61) or E4 (p = 0·65) (S3 Appendix).

Stratification revealed that the trend in full HPSFs was only statistically significant for boys (ES at E4 = -0.24, p = 0.004), whilst in partial HPSFs the trend was only statistically significant for girls (ES at E4 = -0·20, p = 0·002) (S4 Appendix). The sensitivity analysis in which only children were included with four year exposure, resulted in comparable ESs in full (ES = -0·17, B = -0·16 (95%CI -0·27 to -0·06) p = 0·002) and partial HPSFs (ES = -0·13, B = -0·12 (95%CI -0·22

**Table 3. Estimated intervention trends on body composition and estimated intervention effects on PA behaviours and dietary behaviours.**

| | | Full HPSF vs control | | | Partial HPSF vs control | | |
|---|---|---|---|---|---|---|---|
| | | B (95% CI) | P | ES | B (95% CI) | p | ES |
| BMIz (n = 2091) | E1 | -0·043 (-0·067 to -0·019) | **0·000** | -0·04 | -0·039 (-0·063 to -0·016) | **0·001** | -0·04 |
| | E2 | -0·086 (-0·134 to -0·039) | **0·000** | -0·09 | -0·079 (0·126 to −0·031) | **0·001** | -0·08 |
| | E3 | -0·129 (-0·201 to -0·058) | **0·000** | -0·13 | -0·118 (-0·189 to -0·047) | **0·001** | -0·12 |
| | E4 | -0·173 (-0·267 to -0·078) | **0·000** | -0·17 | -0·157 (-0·252 to -0·063) | **0·001** | -0·16 |
| WC (n = 2094) | E1 | -0·364 (-0·582 to -0·146) | **0·001** | -0·06 | -0·390 (-0·606 to -0·173) | **0·000** | -0·06 |
| | E2 | -0·728 (-1·164 to -0·292) | **0·001** | -0·11 | -0·779 (-1·212 to -0·347) | **0·000** | -0·12 |
| | E3 | -1·092 (-1·746 to -0·438) | **0·001** | -0·17 | -1·169 (-1·818 to -0·520) | **0·000** | -0·18 |
| | E4 | -1·456 (-2·328 to -0·584) | **0·001** | -0·22 | -1·559 (-2·424 to -0·693) | **0·000** | -0·24 |
| Sedentary behaviour (n = 1725) | E1 | -0·475 (-1·477 to 0·527) | 0·35 | -0·08 | 0·206 (-0·808 to 1·219) | 0·69 | 0·04 |
| | E4 | 0·606 (-0·750 to 1·962) | 0·38 | 0·11 | 1·422 (0·156 to 2·687) | **0·028** | 0·25 |
| LPA (n = 1725) | E1 | 0·260 (-0·533 to 1·052) | 0·52 | 0·06 | -0·197 (-0·998 to 0·605) | 0·63 | -0·05 |
| | E4 | -0·620 (-1·657 to 0·417) | 0·24 | -0·14 | -1·124 (-2·090 to -0·157) | **0·023** | -0·26 |
| MVPA (n = 1725) | E1 | 0·212 (-0·203 to 0·628) | 0·32 | 0·09 | 0·007 (-0·415 to 0·429) | 0·97 | 0·00 |
| | E4 | 0·004 (-0·554 to 0·563) | 0·99 | 0·00 | -0·270 (-0·793 to 0·252) | 0·31 | -0·11 |
| School water consumption (n = 1840) | E1 | 0·795 (0·609 to 0·981) | **0·000** | 0·68 | | | |
| | E4 | 0·540 (0·334 to 0·747) | **0·000** | 0·46 | | | |
| Healthy dietary behaviour (n = 1527) | E1 | 0·177 (0·032 to 0·323) | **0·017** | 0·18 | | | |
| | E4 | -0·051 (-0·261 to 0·159) | 0·63 | -0·05 | | | |
| Unhealthy dietary behaviour (n = 1516) | E1 | -0·116 (-0·229 to -0·003) | **0·044** | -0·18 | | | |
| | E4 | -0·049 (-0·215 to 0·116) | 0·56 | -0·07 | | | |

Bold p-value = significant (<0·05) difference between conditions.

Abbreviations: B, Beta; BMI, body mass index; CI, confidence interval; ES, effect size; HPSF, Healthy Primary School of the Future; LPA, light physical activity; MVPA, moderate to vigorous physical activity, PA, physical activity, WC, waist circumference.

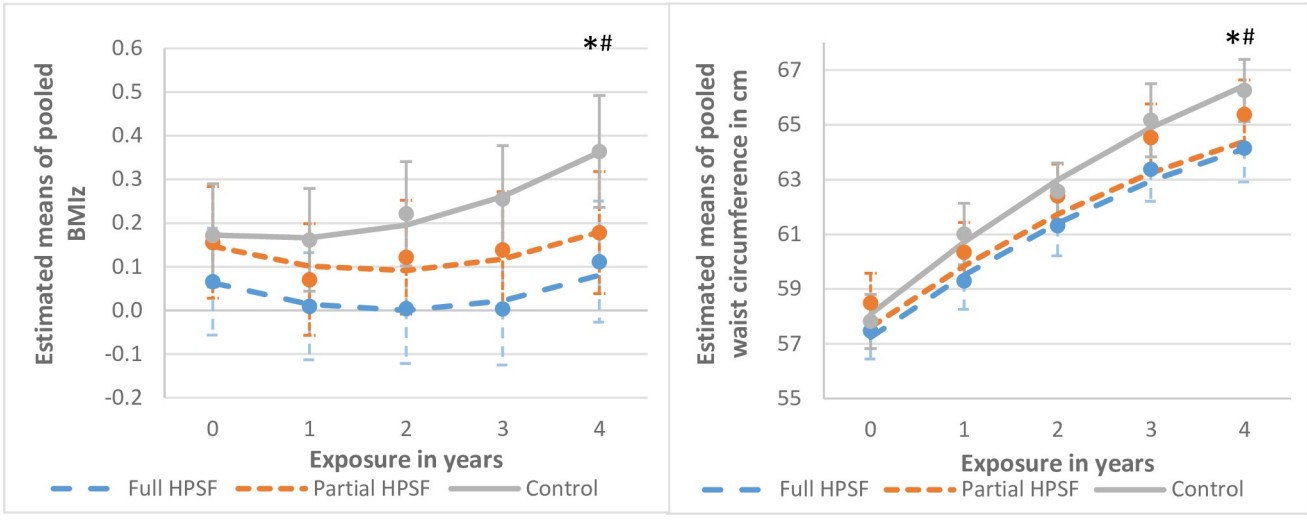

**Fig 2. Estimated means and estimated trends of children's pooled BMIz and waist circumference.** Abbreviations: BMIz, body mass index z-score; HPSF, Healthy Primary School of the Future. * = significant difference in trend over time between Full HPSF and control. # = significant difference in trend over time between Partial HPSF and control.

to -0·02) p = 0·016). The analyses with BMI in kg/m$^2$ resulted in a comparable trend, but effect sizes were greater at E4 with -0·28 for full HPSFs (B = -0·61 (95%CI-0·88 to -0·34), p<0·000) and -0·25 for partial HPSFs compared to control schools (B = -0·54 (95%CI-0·81 to -0·27), p<0·000).

### Secondary outcomes (Tables 3 and 4, Figs 3 and 4, S5 Appendix)

### Waist circumference (Table 3)

The WC of children in all conditions increased over four-year exposure. Compared to control schools, the increase in WC in full HPSFs was significantly less steep, resulting at E4 in a decrease of WC of 1.46 cm (ES -0·22 (p = 0·001)) and in partial HPSFs in a decrease of 1.56 cm (ES -0·24 (p = 0·000)) compared to control schools.

### PA behaviours (Table 3, Fig 3)

All children, regardless of interventions, increased sedentary behaviours, and decreased low PA (LPA) and moderate-vigorous PA (MVPA) behaviours during the study. Changes in PA behaviours (sedentary behaviour, LPA and MVPA) were not significantly different between full HPSFs and control schools at E1 (sedentary ES = -0·08, LPA ES = 0·06, MVPA ES = 0·09) and E4 (sedentary ES = 0·11, LPA ES = -0·14, MVPA ES = 0·00). In partial HPSFs, PA behaviours were not significantly different from control schools at E1 (sedentary ES = 0·04, LPA ES = -0·05, MVPA ES = 0·00), whereas at E4 significantly higher levels of sedentary behaviour (ES = 0·25, p = 0·028) and lower levels of LPA (ES = -0·26, p = 0·023) were detected as compared to control schools.

### Dietary behaviours (Tables 3 and 4, Figs 3 and 4)

In full HPSFs at E1, healthy behaviours increased (ES = 0·18), unhealthy behaviours decreased (ES = -0·18), school water consumption increased (ES = 0·68), and the intake of the majority

**Table 4. Estimated intervention effects at E1 and E4 on lunch intake.**

| | | Full HPSF vs control | |
|---|---|---|---|
| | | OR (95% CI) | p |
| Fruit (% yes) (n = 2032) | E1 | 1·740 (1·191 to 2·541) | **0·004** |
| | E4 | 0·432 (0·267 to 0·698) | **0·001** |
| Vegetables (% yes) (n = 2034) | E1 | 2·575 (1·707 to 3·888) | **0·000** |
| | E4 | 1·605 (0·970 to 2·656) | 0·065 |
| Grains (% yes) (n = 2033) | E1 | 0·535 (0·291 to 0·983) | **0·044** |
| | E4 | 0·921 (0·433 to 1·962) | 0·832 |
| Dairy (% yes) (n = 2033) | E1 | 3·040 (2·074 to 4·461) | **0·000** |
| | E4 | 1·647 (1·031 to 2·632) | **0·037** |
| Water (% yes) (n = 2031) | E1 | 1·029 (0·705 to 1·503) | 0·882 |
| | E4 | 1·132 (0·715 to 1·790) | 0·597 |
| Butter (% yes) (n = 2027) | E1 | 0·250 (0·174 to 0·359) | **0·000** |
| | E4 | 0·283 (0·176 to 0·454) | **0·000** |
| Minimum two healthy food groups at lunch (% yes) (n = 2037) | E1 | 3·037 (1·644 to 5·611) | **0·000** |
| | E4 | 1·289 (0·654 to 2·543) | 0·463 |

Bold p-value = significant (≤0·05) difference between conditions.

Abbreviations: CI, confidence interval; OR, odds ratio; HPSF, Healthy Primary School of the Future.

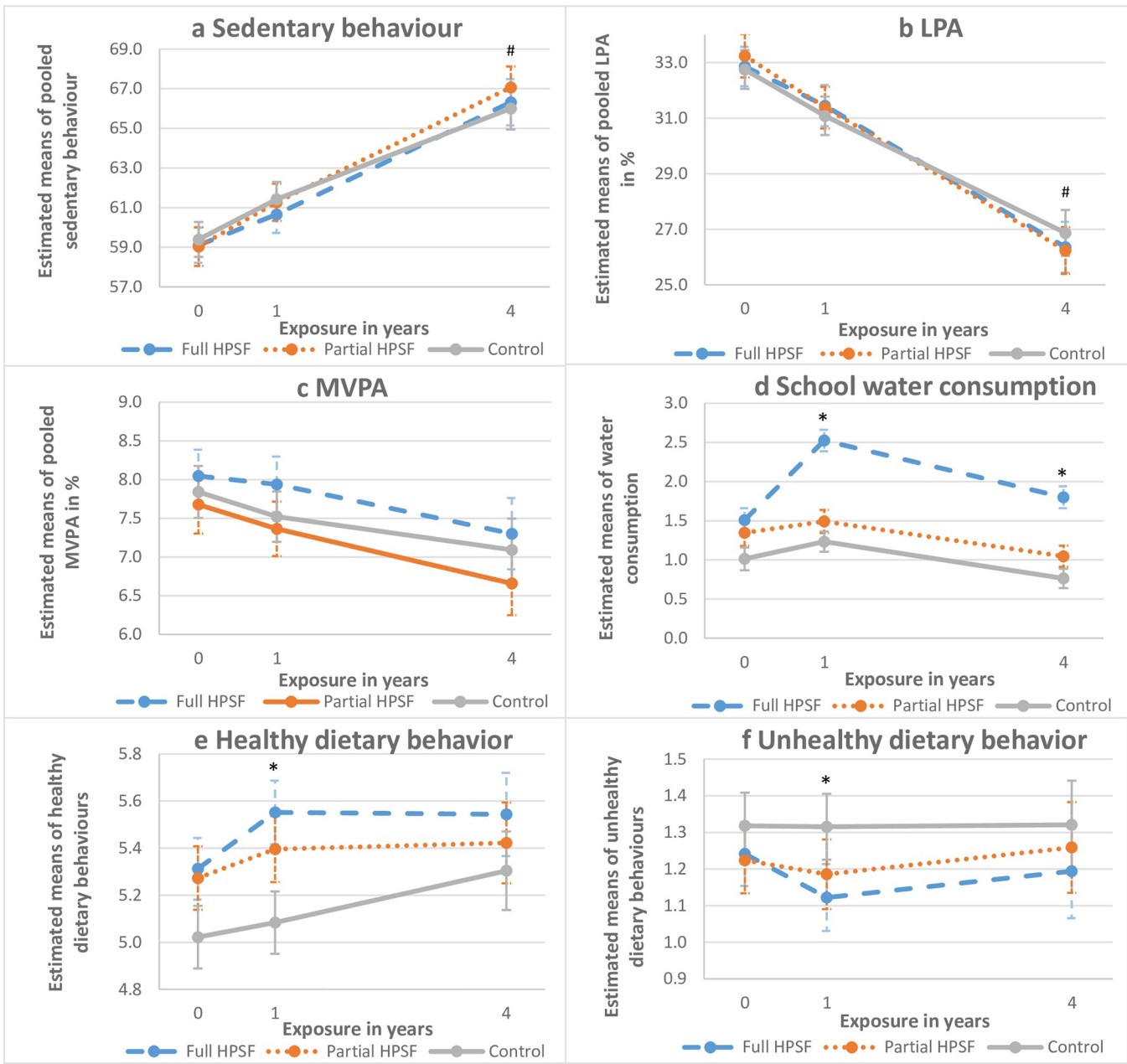

**Fig 3.** Estimated means of children's dietary and PA behaviours at baseline, one and four years of exposure; a) Sedentary behaviour; b) LPA; c) MVPA; d) school water consumption; e) Healthy dietary behaviour; f) Unhealthy dietary behaviour. Abbreviations: HPSF, Healthy Primary School of the Future; LPA, light physical activity; MVPA, moderate to vigorous physical activity. * = significant difference between Full HPSF and control. # = significant difference between Partial HPSF and control.

of healthy lunch components increased compared to control schools (vegetable (OR = 2·58), fruit (OR = 1·74), dairy (OR = 3·04), intake of a minimum of two healthy food groups (OR = 3·04)). Butter consumption (OR = 0·25) and grain intake during lunch (OR = 0·54) decreased significantly. At E4, most of the short-term intervention effects in full HPSFs were no longer significant (healthy behaviours ES = -0·05; unhealthy behaviours ES = -0·07). The intervention effects on school water consumption (ES = 0·46) and dairy intake remained

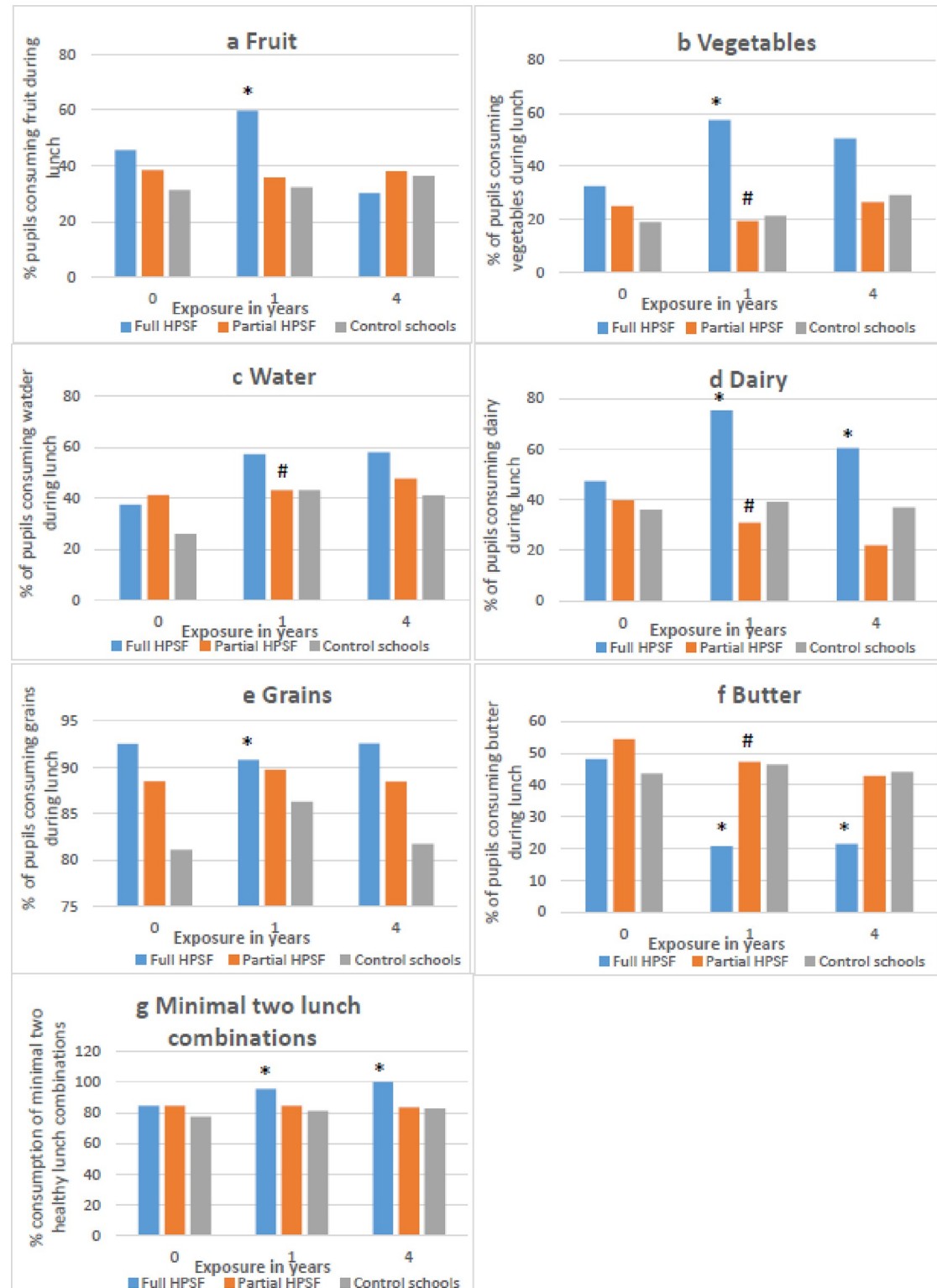

**Fig 4.** Observed percentage of lunch consumption: a) fruit, b) vegetables, c) water, d) dairy, e) grains, f) butter, h) minimal two healthy lunch combinations at baseline, one and four years of exposure. Abbreviations: BMI, body mass index; HPSF, Healthy Primary School of the Future. * = Significant difference between Full HPSF and control. # = Significant difference between Partial HPSF and control.

significant (OR = 1·64). A significant decrease was seen in fruit intake during lunch (fruit OR = 0·43) between E1 and E4.

Annual estimated intervention effects on PA and dietary behaviours variables are given in S5 Appendix.

## Discussion

In the current quasi-experimental trial, we studied the effects of the HPSF school-based health-promoting interventions on BMIz, WC, PA, and dietary behaviours. Both full and partial HPSFs resulted in favourable changes on BMIz and WC, whereby longer intervention exposure resulted in higher effect sizes.

The results are in line with our initial hypothesis, which described increasing intervention effects on BMIz and WC with increasing intervention exposure. The HPSF intervention mostly resulted in the preservation of children's body composition, and did not lead to major changes in weight status. By contrast, in control schools, a continuous worsening of BMIz and WC is visible. Although the effect sizes after four years on body composition are considered small [21], they are slightly higher than reported in existing literature [7, 8, 22, 23]. The number of studies measuring longitudinal effects on both BMIz and WC in a large group of children is very limited. Although there is no academic consensus on the minimal relevant difference of BMIz, previous studies showed that changes in BMIz as small as 0.1 in children might already have beneficial public health consequences, as achieved in the current study [24, 25]. What strengthens the validity of our findings are the simultaneous improvements in both BMIz and WC. WC is known to have a stronger association with cardiometabolic health in children than BMI, as improvements in WC indicate a decrease in fat mass [26]. The ongoing negative trend in control schools implies that not intervening will lead to a growth of the obesity epidemic, with lifelong complications for children's health and wellbeing.

In full HPSFs, the majority of dietary behaviours improved after one year, but did not further improve or even relapsed slightly after four years. By far the highest effect sizes were found on school water consumption. Water consumption was, together with dairy consumption, the only dietary behaviour which remained improved compared to control schools in the long term. The impact of this long-term change in dietary pattern might be of high societal relevance, as the consumption of sugar-sweetened beverages is linked to excessive weight gain [27]. It is noteworthy that some dietary behaviours were maintained in full HPSFs, yet were no longer significantly different from control schools due to a change of dietary behaviours in control schools. This could possibly be attributed to the participation of three control schools with the European School Fruit and Vegetables Scheme starting between E3 and E4. In contrast to dietary behaviours, PA behaviours remained unchanged over time in both full and partial HPSFs compared to control schools, with some unfavourable changes in partial HPSFs after four years. Possibly, the PA interventions in partial control schools did not intrinsically motivate children sufficiently to engage in PA. In addition, it is possible that children compensated PA behaviours at school with more sedentary behaviours at home. PA behaviours deteriorated over time in all school types, as is common from a life course perspective in a study population reaching puberty [28]. The lack of long-term effectiveness on dietary and PA behaviours is not at all surprising, as there is a limit to the number of behaviours a child can and wants to adapt. In addition, the intervention was not designed to achieve a continuous improvement in dietary and PA behaviours.

It is striking that the long-term favourable intervention effects on dietary and PA behaviours are mostly absent, whilst favourable intervention effects on BMI and WC increase over time compared to the control situation. From a homeostatic perspective, it is unlikely that

longitudinal changes in BMI occurred without longitudinal changes in dietary and PA behaviours. It can be questioned whether our lifestyle measurement instruments were sensitive enough to measure the often subtle changes in the co-existence and interaction of several dietary and PA behaviours which lead to longitudinal changes in energy balance.

Our study emphasises the importance of longitudinal study designs in school health-promoting interventions. In general, it is assumed that longer interventions allow for repeated exposure and provide more opportunity for changing social norms, i.e. water and dairy consumption instead of sugar-sweetened beverages, and consequently higher effectiveness [29]. However, this is not uniformly confirmed in literature, partly because the number of school-based interventions studies with a longitudinal duration and sufficient power is very limited (S1 Appendix) [22]. Future research is necessary to study the course of prolonged (>four years) exposure of school-based interventions on BMI and healthy behaviours, as this information further strengthens evidence for public policy makers to sustainably implement health-promoting interventions in the school context.

Decisions on implementation of school-based health promoting interventions should be based on the interplay between longitudinal societal benefits and revenues, including health behaviours, costs, public support, food waste, educational achievements, and well-being of students [30]. Several of these multi-domain benefits have been studied for HPSF [12]. In both partial and full HPSFs, fewer conflicts were reported at school, whereby it was only in full HPSFs that teachers observed an increase in social behaviour [31]. Caregivers and children are generally satisfied with both partial and full HPSFs [30]. The short-term societal benefits have not yet achieved sufficient financial returns in order to compensate the costs. Depending on the value that stakeholders attach to non-financial benefits, a full HPSF can, however, be a cost-effective investment [30]. Long-term modelling of HPSF effects on BMIz into adulthood showed that an HPSF is a cost-effective and equitable strategy for combatting the lifetime burden of unhealthy lifestyles, given that the improvements in BMIz will be maintained over time [17]. Although full HPSF is considered by most schools as an invasive intervention which requires high investments, there seems to be public support to implement full HPSF; currently, several other schools in the Netherlands are implementing full HPSF.

With over 2000 participating children, the large sample size is a strength of this study. Although attrition caused by the dynamic open cohort design of our study (e.g., graduation of students before T4, and enrolment after T0) was higher than anticipated in our original power calculation, we included more participants than originally calculated [12]. Also, attrition due to drop out was low, with only 53 children actively stopping participation over a four-year period. Another strength of the study is the objective methods used to assess body composition and PA. The subjective measurements to assess dietary behaviour might have led to socially desirable answers. The questionnaires completed by children resulted in a high average annual response rate varying between 92·1% and 95·0%. The parental questionnaire used to determine covariates had a lower response rate (46.0%), yet allowed us to ask more questions in detail. By using exposure as a fixed factor in our analyses, we were able to include all children who enrolled and left school during the study period whilst accounting for the varying exposure to the school condition, and thereby tackled a common disadvantage of dynamic cohort designs. Due to the quasi-experimental design of this study, we were able to enrol schools based on motivation. This reflects the real-life situation of school health promotion and increases ecological validity. The lack of randomization resulted in baseline differences, with children in control schools showing more unfavourable dietary and PA behaviours and a higher BMI. This could have led to an underestimation of effects, as there was less room for improvement in intervention schools. To deal with the lack of randomization, we controlled in all analyses

for sex, study year at E0, socioeconomic status score, ethnicity, and BMIz at E0 (only for the behavioural outcomes).

Schools are complex settings to implement health-promoting interventions as many stakeholders are involved, and both population and school characteristics vary highly. Consequently, similar interventions do not automatically lead to similar outcomes across schools [32]. In an earlier study, we found our study population to be a good representation of the region, but external validation with a national sample was moderate [16]. Children in our sample more often had a Western ethnicity and a relatively low socioeconomic status. Overweight and obesity prevalence at 18·9% is higher than the national average of 15–17% [19]. Therefore, we recommend carefully studying implementation and effectiveness of the current intervention in various school contexts with higher socioeconomic statuses and more ethnically diverse populations.

In conclusion, the current interventions proved effective in preventing weight gain expressed in BMIz and WC in children. Longer exposure resulted in increasing effect sizes on body composition. Simultaneously, longer exposure enables a shift in society's norms of behaviour, which could eventually result in a high impact at the population level [29]. Without interventions, it is most likely that the obesity epidemic will worsen. From a societal perspective, it is not only of vital importance to bring the obesity epidemic to a halt, but also to reverse it. If school-based interventions can be combined with individualised interventions targeted at children in the risk category, and by introducing interventions in the wider societal contexts, including the media and food industry, a structural change in the ongoing worrisome obesity epidemic is possible [7, 29].

## Supporting information

**S1 Appendix.**
(DOCX)

**S2 Appendix.**
(DOCX)

**S3 Appendix.**
(DOCX)

**S4 Appendix.**
(DOCX)

**S5 Appendix.**
(DOCX)

## Acknowledgments

We thank the trial participants and their parents or guardians for agreeing to take part in the trial, the participating schools, and all the trial staff.

## Author Contributions

**Conceptualization:** M. Willeboordse, N. H. M. Bartelink, P. van Assema, S. P. J. Kremers, H. H. C. M. Savelberg, M. T. H. Hahnraths, L. Vonk, M. Oosterhoff, C. P. van Schayck, M. W. J. Jansen.

**Formal analysis:** M. Willeboordse, N. H. M. Bartelink, B. Winkens.

**Funding acquisition:** M. Willeboordse, C. P. van Schayck, M. W. J. Jansen.

**Investigation:** M. Willeboordse, N. H. M. Bartelink, M. T. H. Hahnraths, L. Vonk, M. Oosterhoff.

**Project administration:** M. Willeboordse.

**Supervision:** M. Willeboordse, C. P. van Schayck, M. W. J. Jansen.

**Writing – original draft:** M. Willeboordse.

**Writing – review & editing:** N. H. M. Bartelink, P. van Assema, S. P. J. Kremers, H. H. C. M. Savelberg, M. T. H. Hahnraths, L. Vonk, M. Oosterhoff, C. P. van Schayck, B. Winkens, M. W. J. Jansen.

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
