## [Decision Letter · Decision Letter 0]

24 Mar 2022

PONE-D-21-31405Battling the obesity epidemic with a school-based intervention: Long-term effects of a quasi-experimental studyPLOS ONE

Dear Dr. Willeboordse,

Thank you for submitting your manuscript to PLOS ONE. After careful consideration, we feel that it has merit but does not fully meet PLOS ONE’s publication criteria as it currently stands. Therefore, we invite you to submit a revised version of the manuscript that addresses the points raised during the review process. I apologize again for the delay in coming to a decision. The good news is that the reviewers see merit in your work. They came up with some comprehensible questions that mainly are requests for additional information, which you should be able to answer satisfactorily.

We look forward to receiving your revised manuscript.

Kind regards,

Clemens Fürnsinn, Ph.D.

Academic Editor

PLOS ONE

Journal Requirements:

2. Thank you for stating the following in the Acknowledgments/ Funding Section of your manuscript: 

This study was funded by the Limburg provincial authorities (www.limburg.nl), Project Number 200130003 (received by CvS), by Maastricht University (www.maastrichtuniversity.nl), Project Number 200130003 (received by MJ) and by FrieslandCampina (www.frieslandcampina.nl) (received by MW), Project Number LLMV00. The funders had no role in study design, data collection and analysis, decision to publish, or preparation of the manuscript.

This study was funded by the Limburg provincial authorities (www.limburg.nl), Project Number 200130003 (received by CvS), by Maastricht University (www.maastrichtuniversity.nl), Project Number 200130003 (received by MJ) and by FrieslandCampina (www.frieslandcampina.nl) (received by MW), Project Number LLMV00. The funders had no role in study design, data collection and analysis, decision to publish, or preparation of the manuscript.

3. We noted in your submission details that a portion of your manuscript may have been presented or published elsewhere. Please clarify whether this publication was peer-reviewed and formally published. If this work was previously peer-reviewed and published, in the cover letter please provide the reason that this work does not constitute dual publication and should be included in the current manuscript.

6. You indicated that ethical approval was not necessary for your study. We understand that the framework for ethical oversight requirements for studies of this type may differ depending on the setting and we would appreciate some further clarification regarding your research. Could you please provide further details on why your study is exempt from the need for approval and confirmation from your institutional review board or research ethics committee (e.g., in the form of a letter or email correspondence) that ethics review was not necessary for this study? Please include a copy of the correspondence as an ""Other"" file.

Reviewers' comments:

Reviewer's Responses to Questions

**Comments to the Author**

1. Is the manuscript technically sound, and do the data support the conclusions?

Reviewer #1: Yes

Reviewer #2: Partly

2. Has the statistical analysis been performed appropriately and rigorously? 

Reviewer #1: N/A

Reviewer #2: Yes

3. Have the authors made all data underlying the findings in their manuscript fully available?

Reviewer #1: Yes

Reviewer #2: Yes

4. Is the manuscript presented in an intelligible fashion and written in standard English?

Reviewer #1: Yes

Reviewer #2: Yes

5. Review Comments to the Author

Reviewer #1: The presented study reports on a real life intervention in schools. The results are important and highly valuable to policy maker, also in other countries than The Netherlands.

The authors have adressed all questions of other reviewers. Nothing to add. The only remark I want to make - it might be of interest for the non-Dutch reader to learn more about the the school environment for example - are materials from food companies as teaching materials used/allowed, participation in the school milk school fruit Programm ...

I agree with the authors, we need more of this kind of evaluated interventions.

Reviewer #2: Review Comments

PONE-D-21-31405 Battling the obesity epidemic with a school-based intervention: Long-term effects of a quasi-experimental study

Major comments:

1. Power calculations justifying sample size are not presented, as well as justification for the effect size included in the calculation. Have they been published elsewhere? A brief description in this paper would be helpful.

2. The rate of attrition (drop out) could be clarified. What was the rate of attrition by the study arms? What was the attrition rate assumed for power calculations? Is attrition randomly distributed? If there is non-random attrition, what statistical techniques are used to identify and adjust for the attrition bias?

3. Any comments on spill-overs? How is the contamination controlled for (e.g. the participation of three comparison schools with the European School Fruit and Vegetables Scheme)?

4. For a combined table on baseline characteristics, the last column usually gives the p value for the comparison between study groups. Why is the p value on baseline differences not presented and only the mean difference is reported?

5. Can the authors report both adjusted and unadjusted models, to avoid reporting bias? Do both adjusted and unadjusted analyses yield the same results?

6. “Comparison schools” instead of “control schools” as this is not an RCT?

7. Can the authors comment on food waste in this 4-year intervention? Food waste in school nutrition programs has received increasing attention.

Minor comments:

Introduction

8. Reference is missing for this statement: Current evidence suggests that school-based interventions show small favourable effects in terms of body mass index (BMI), and dietary and physical activity (PA) behaviours.

9. For L109-L113, is there any particular reason why result on BMIz in the short term evaluation is not reported? It would appear that effects on BMIz only started to show after two years, while effects on BMIz was significant from E1 to E4 as reported in the Results section. This could be clarified.

10. The statement “We hypothesise that improvements in dietary and PA behaviours will improve after one year of exposure…” (L117) could be clarified. Does it refer to an increasing intervention effect?

Study design

11. The numbers reported in L153-L157 are not lined up with numbers on E0-E4 in Participation expressed in exposure (in years) (Figure 1). This could be clarified.

Results

12. Was the stratification by gender specified in the pre-analysis plan? (L269-L270)

13. Results on PA behaviors at E4 appears to be counter-intuitive (L294-L295). This could be commented on in the Discussion section.

14. How are Figure 3 & Figure 4 referenced in the text?

Discussion

15. On L390, “water and dairy consumption instead of soft drinks” is stated. Is a decrease in soft drinks detected to support the statement? This could be reported as in the current study, the greatest effect was found in children’s water consumption.

6. PLOS authors have the option to publish the peer review history of their article (what does this mean?). If published, this will include your full peer review and any attached files.

Reviewer #1: No

Reviewer #2: No

---

## [Author Response · Author response to Decision Letter 0]

13 Apr 2022

We thank the reviewers for their comments on our manuscript. Based on the comments, we were able to further improve our manuscript. All changes made when revising the manuscript have been highlighted in the revised manuscript. In our response to the reviewers, we provide a point-by-point response to the reviewers’ comments.

---

## [Editor Report · Decision Letter 1]

18 Jul 2022

Battling the obesity epidemic with a school-based intervention: Long-term effects of a quasi-experimental study

PONE-D-21-31405R1

Dear Dr. Willeboordse,

We’re pleased to inform you that your manuscript has been judged scientifically suitable for publication and will be formally accepted for publication once it meets all outstanding technical requirements.

Kind regards,

Clemens Fürnsinn, Ph.D.

Academic Editor

PLOS ONE
---

## [Editor Report · Acceptance letter]

2 Sep 2022

PONE-D-21-31405R1 

Battling the obesity epidemic with a school-based intervention: Long-term effects of a quasi-experimental study 

Dear Dr. Willeboordse:

I'm pleased to inform you that your manuscript has been deemed suitable for publication in PLOS ONE. Congratulations! Your manuscript is now with our production department. 

Kind regards, 

on behalf of

Prof. Dr. Clemens Fürnsinn 

Academic Editor

PLOS ONE